# Approximation of Glomerular Filtration Rate after 1 Year Using Annual Medical Examination Data

**DOI:** 10.3390/jcm13144207

**Published:** 2024-07-18

**Authors:** Keiji Hirai, Taisuke Kitano, Keiji Nakayama, Fujiko Morita, Hajime Satomura, Takahisa Tanaka, Toru Yoshioka, Masahiko Matsumoto, Yuichi Kimura, Taku Shikanai, Koji Sasaki, Zhiying Zhang, Kiyonori Ito, Susumu Ookawara, Yoshiyuki Morishita

**Affiliations:** 1Division of Nephrology, First Department of Integrated Medicine, Saitama Medical Center, Jichi Medical University, Saitama 330-8503, Japan; lovbrace@yahoo.co.jp (T.K.); kiyonori.ito@gmail.com (K.I.); ookawaras0123@gmail.com (S.O.); ymori@jichi.ac.jp (Y.M.); 2Omiya Medical Association Cohort Study Group, Saitama 331-8689, Japan; ay9k-nkym@asahi-net.or.jp (K.N.); moritafujiko2@gmail.com (F.M.); m010004545@yahoo.co.jp (H.S.); tanaka.dmclinic@gmail.com (T.T.); tyoshioka0907@gmail.com (T.Y.); masiko@plum.plala.or.jp (M.M.); 3Nakayama Clinic, Saitama 330-0855, Japan; 4Morita Clinic, Saitama 337-0051, Japan; 5Satomura Clinic, Saitama 331-0813, Japan; 6Tanaka Diabetes Clinic Omiya, Saitama 330-0846, Japan; 7Yoshioka Clinic, Saitama 330-0851, Japan; 8Matsumoto Clinic, Saitama 331-0822, Japan; 9LIMNO Co., Ltd., Tottori 680-8634, Japan; kimura.yuichi@outlook.com; 10BioICT Co., Ltd., Yokohama 227-0038, Japan; 11Azest, Inc., Chiyoda 101-0052, Japan; shikanai@k.azest.co.jp (T.S.); sasaki@azest.co.jp (K.S.); zhang@azest.co.jp (Z.Z.)

**Keywords:** estimated glomerular filtration rate, annual change, chronic kidney disease, hemoglobin, uric acid

## Abstract

**Background**: This cohort study was conducted to devise an approximation formula for predicting the glomerular filtration rate (GFR) after 1 year using annual medical examination data from the general population. **Methods**: Consecutive annual medical examination data were obtained for 41,337 inhabitants. Machine learning with the random forest method was used to assess the importance of each clinical parameter in terms of its association with estimated GFR (eGFR) after 1 year. An approximation formula was developed by multiple linear regression analysis based on the four most important clinical parameters. The relationship between the GFR after 1 year approximated by our formula and the eGFR after 1 year was analyzed using Pearson’s correlation coefficient. **Results**: The following approximation formula was obtained by multiple linear regression analysis: approximate GFR after 1 year (mL/min/1.73 m^2^) = −0.054 × age + 0.162 × hemoglobin − 0.085 × uric acid + 0.849 × eGFR + 11.5. The approximate GFR after 1 year was significantly and strongly correlated with the eGFR at that time (r = 0.884; *p* < 0.001). **Conclusions**: An approximation formula including age, hemoglobin, uric acid, and eGFR may be useful for predicting GFR after 1 year among members of the general population.

## 1. Introduction

Chronic kidney disease (CKD) is one of the most common diseases worldwide; its incidence and prevalence continue to increase each year [1]. CKD progression is associated with increased risks of mortality and cardiovascular events [2,3]. Therefore, early identification of individuals with a high risk of renal function decline is important for early interventions to prevent CKD progression.

Numerous factors have been reported to be associated with renal function decline, namely male sex, age, smoking, body mass index (BMI), systolic blood pressure (SBP), diastolic blood pressure (DBP), hemoglobin, uric acid, triglyceride, high-density lipoprotein cholesterol (HDL-C), low-density lipoprotein cholesterol (LDL-C), hemoglobin A1c, urinary protein, and estimated glomerular filtration rate (eGFR) (Table 1) [4,5,6,7,8,9,10,11,12,13,14,15,16,17,18,19,20,21,22]. In the previous studies that investigated the factors associated with renal function decline, the clinical outcomes examined were a decline in eGFR, progression of CKD, and progression to end-stage renal disease [4,5,6,7,8,9,10,11,12,13,14,15,16,17,18,19,20,21,22]. However, an approximation formula for the prediction of future renal function derived from risk factors associated with renal function decline has not yet been established. Therefore, in this study, we developed an approximation formula for predicting the GFR after 1 year using annual medical examination data from the general population of Japan.

## 2. Methods

### 2.1. Ethical Approval

This study was approved by the ethics committee of the Omiya Medical Association on 1 April 2016 (approval number: 2016001) and performed in accordance with ethical principles outlined in the Declaration of Helsinki. The ethics board waived the requirement for informed consent because of the retrospective nature of the study. Therefore, information regarding this study was displayed on notice boards in relevant institutions to inform all participants of their right to opt out.

### 2.2. Study Participants

We collected annual medical examination data of residents in Saitama City, Japan, from 2011 to 2019. All examinations were performed at primary care clinics and community hospitals in Saitama City. The inclusion criteria were age ≥20 years and participation in an annual medical examination for ≥2 consecutive years from 2011. The exclusion criteria were hemodialysis, peritoneal dialysis, and renal transplantation.

### 2.3. Study Design

This investigation was a retrospective population-based cohort study that utilized annual medical examination data for residents in Saitama City, Japan. Consecutive annual medical examination data from 2011 to 2019 were obtained by retrospective review of a medical database provided by Saitama City. The importance of each clinical parameter in terms of its association with the eGFR after 1 year was assessed by machine learning using the random forest method. An approximation formula for predicting GFR after 1 year was developed by multiple linear regression analysis based on the four most important clinical parameters. The relationship between the GFR after 1 year approximated by our formula and the eGFR after 1 year was analyzed using Pearson’s correlation coefficient. Agreement between the approximate GFR after 1 year and the eGFR after 1 year was analyzed using Bland–Altman analysis.

### 2.4. Laboratory Methods

Blood and urine parameters were measured by commercial or hospital laboratories. eGFR was calculated using a modified version of the Modification of Diet in Renal Disease formula from the Japanese Society of Nephrology: eGFR (mL/min/1.73 m^2^) = 194 × age^−0.287^ × serum creatinine^−1.094^ (multiplied by 0.739 for women) [23]. Blood pressure was measured with the participant in a sitting position at rest, using an automated upper arm cuff. Annual changes in clinical parameters were determined by subtracting the values from the values after 1 year.

### 2.5. Statistical Analyses

Data processing, machine learning implementation, and statistical analyses were performed using the KNIME Analytics Platform version 4.7.1 (KNIME, Zurich, Switzerland). We selected the clinical parameters that were available in the annual medical examination dataset of residents in Saitama City and were reported to be associated with renal function decline in previous reports. The final dataset included the following 14 variables: male sex, age, smoking, BMI, SBP, DBP, hemoglobin, uric acid, triglyceride, HDL-C, LDL-C, hemoglobin A1c, urinary protein, and eGFR; all variables were reportedly associated with renal function decline in previous studies (Table 1) [4,5,6,7,8,9,10,11,12,13,14,15,16,17,18,19,20,21,22]. Urinary protein was expressed in a semi-quantitative manner—grade 1, −; grade 2, ±; grade 3, 1+; grade 4, 2+; grade 5, ≥3+—because urinalysis was performed with a urine dipstick test in annual medical examination. Data were expressed as means ± standard deviations for continuous variables, and as counts and percentages for categorical variables. Data were normalized and divided into a training set and test set. We used 80% of the data for training and the remaining 20% to validate the results. Afterward, the importance of each parameter was assessed using a machine learning-based random forest method, sorted in descending order, and illustrated using a bar chart. Next, test data were applied to the prediction model and the accuracy of the model was evaluated (Figure 1). The first dataset included 14 independent variables: sex, age, smoking history, BMI, SBP, DBP, hemoglobin, uric acid, triglyceride, HDL-C, LDL-C, hemoglobin A1c, urinary protein, and eGFR. The second dataset included 14 independent variables: sex, age, annual change in smoking history, annual change in BMI, annual change in SBP, annual change in DBP, annual change in hemoglobin, annual change in uric acid, annual change in triglyceride, annual change in HDL-C, annual change in LDL-C, annual change in hemoglobin A1c, annual change in urinary protein, and eGFR. Annual changes in sex, age, and eGFR were not used as independent variables because sex cannot be changed, the annual change in age was 1 year for all participants, and annual change in eGFR was the dependent variable. Based on a previous report that showed that four-variable and eight-variable equations are similar with respect to predictive accuracy [24], the four most important parameters in terms of associations with eGFR after 1 year, according to the random forest method, were included in multiple linear regression analysis. Data were divided into a training set and test set. We used 80% of the data for training and the remaining 20% to validate the results. Next, an approximation formula was established by multiple linear regression analysis using the forced entry method. After test data were applied to the approximation formula, missing values were excluded and the accuracy of the formula was evaluated (Figure 2). We performed two sets of multiple linear regression analyses: one focused on clinical parameters and the other focused on annual changes in clinical parameters. Correlation between the approximate GFR after 1 year and the eGFR after 1 year was evaluated using Pearson’s correlation coefficient. Agreement between the approximate GFR after 1 year and the eGFR after 1 year was assessed using Bland–Altman analysis. *p* values < 0.05 were considered statistically significant.

Data were normalized (Step 1) and divided into a training set and test set (Step 2). Afterward, the importance of each parameter was assessed using the random forest method (Step 3), sorted in descending order (Step 4), and illustrated using a bar chart (Step 5). Next, test data were applied to the prediction model (Step 6) and the accuracy of the model was evaluated (Step 7).

Data were divided into a training set and test set (Step 1). Next, an approximation formula was established by multiple linear regression analysis using the forced entry method (Step 2). After test data were applied to the approximation formula (Step 3), missing values were excluded (Step 4) and the accuracy of the formula was evaluated (Step 5).

## 3. Results

### 3.1. Participant Characteristics

The baseline characteristics of the study participants are summarized in Table 2. In total, 349,050 records were obtained for 41,337 participants (16,918 men, 24,419 women; mean age: 64.0 ± 6.9 years; BMI: 22.8 ± 3.2 kg/m^2^). Four thousand five hundred and sixty-one participants (11.0%) had a past or current history of smoking. The mean SBP and DBP were 128.2 ± 16.2 mmHg and 76.3 ± 10.6 mmHg, respectively. The mean hemoglobin level was 13.8 ± 1.3 g/dL, the mean uric acid level was 5.2 ± 1.3 mg/dL, the mean triglyceride level was 108.9 ± 68.0 mg/dL, the mean HDL-C level was 63.8 ± 15.8 mg/dL, the mean LDL-C level was 126.2 ± 30.0 mg/dL, and the mean hemoglobin A1c level was 5.9 ± 0.6%. Urinary protein grades were −, 86.0%; ±, 8.8%; 1+, 4.0%; 2+, 1.0%; and ≥3+, 0.2%. Figure 3 shows the eGFR trend during the study period. The mean eGFR at baseline was 74.3 ± 14.2 mL/min/1.73 m^2^, and eGFR gradually declined during the study period.

### 3.2. Importance of Clinical Parameters in Terms of Associations with eGFR after 1 Year

Figure 4 shows the importance of each parameter in terms of its association with the eGFR after 1 year. We performed multiple linear regression analysis using the four most important parameters according to the random forest method. This analysis revealed that age (coefficient (β) = −0.054, *p* < 0.001), hemoglobin (β = 0.162, *p* < 0.001), uric acid (β = −0.085, *p* < 0.001), and eGFR (β = 0.849, *p* < 0.001) were independently correlated with the eGFR after 1 year (Table 3 (A)).

We also performed multiple linear regression analysis using the four most important annual changes in parameters according to the random forest method. This analysis revealed that age (β = −0.050, *p* < 0.001), annual change in hemoglobin (β = −0.398, *p* < 0.001), annual change in uric acid (β = −3.205, *p* < 0.001), and eGFR (β = 0.864, *p* < 0.001) were independently correlated with the eGFR after 1 year (Table 3 (B)).

### 3.3. Formula for Approximation of GFR after 1 Year

An approximation formula was developed using variables that showed a significant correlation with the eGFR after 1 year according to multiple linear regression analysis. We developed two formulas: one focused on clinical parameters and the other focused on annual changes in clinical parameters.
Approximate GFR after 1 year (mL/min/1.73 m^2^) = −0.054 × age + 0.162 × hemoglobin − 0.085 × uric acid + 0.849 × eGFR + 11.5(1)
Approximate GFR after 1 year (mL/min/1.73 m^2^) = − 0.050 × age − 0.398 × annual change in hemoglobin − 3.205 × annual change in uric acid + 0.864 × eGFR + 11.9(2)

### 3.4. Correlation between the Approximate GFR after 1 Year and the eGFR after 1 Year

The approximate GFR after 1 year calculated by our formula with clinical parameters (Formula (1)) was significantly and strongly correlated with the eGFR at that time (r = 0.884; *p* < 0.001). The approximate GFR after 1 year calculated by our formula with annual changes in clinical parameters (Formula (2)) was also significantly and strongly correlated with the eGFR at that time (r = 0.894; *p* < 0.001).

### 3.5. Agreement between the Approximate GFR after 1 Year and the eGFR after 1 Year

Bland–Altman analysis showed a moderate agreement between approximate GFR after 1 year calculated by our formula with clinical parameters and eGFR at that time. In total, 97.7% of the points were included within the mean difference ± 1.96 standard deviation (0.0 ± 12.8) (Figure 5). This analysis also showed a moderate agreement between approximate GFR after 1 year calculated by our formula with annual changes in clinical parameters and eGFR at that time. In total, 97.6% of the points were included within the mean difference ± 1.96 standard deviation (0.0 ± 12.1).

## 4. Discussion

In the present study, we found that age, hemoglobin, uric acid, and eGFR were associated with the eGFR after 1 year among members of the general population in Japan. We also found that age, annual change in hemoglobin, annual change in uric acid, and eGFR were associated with the eGFR after 1 year in this population. These findings enabled us to develop an approximation formula for predicting GFR after 1 year; the approximate GFR calculated by this formula was strongly correlated with the eGFR after 1 year. However, as shown in Figure 5, variation between approximate GFR after 1 year and eGFR after 1 year became greater as eGFR after 1 year became higher. It has been reported that variation between GFR measured by inulin clearance and eGFR calculated by the Modification of Diet in Renal Disease formula became greater as the GFR became higher [23]. These findings suggest that the development of an approximation formula for predicting the GFR after 1 year might be challenging among members of the general population.

A higher serum uric acid concentration is reportedly associated with a more rapid decline in creatinine clearance among patients with CKD G1-4 [25]. An observational study showed that an increased serum uric acid concentration was associated with a higher incidence of end-stage kidney disease among patients with CKD G3-4 [26]. Several observational studies involving members of the general population showed that a change in the serum uric acid concentration was negatively associated with a change in the eGFR [15,27]. In the present study, the eGFR after 1 year was negatively associated with the serum uric acid concentration and the annual change in serum uric acid concentration among members of the general population. These results suggest that the serum uric acid concentration is negatively associated with changes in renal function in the population both with and without CKD.

There is evidence that the eGFR declines more rapidly as the hemoglobin concentration decreases among patients with CKD G2-5 [13]. In an observational study, a decreased hemoglobin concentration was associated with a higher incidence of end-stage kidney disease among patients with CKD G4-5 [28]. Several observational studies of healthy individuals showed that a lower hemoglobin concentration was associated with more rapid eGFR decline [12,29]. In the present study, the hemoglobin concentration was positively associated with the eGFR after 1 year among members of the general population. These findings suggest that the hemoglobin concentration is positively associated with changes in renal function in the population both with and without CKD. However, the present study also showed that the annual change in hemoglobin concentration was negatively associated with the eGFR after 1 year. Mild renal dysfunction is suspected to enhance renal erythropoietin production in early-stage CKD [30]. A longitudinal cohort study revealed that eGFR decline was associated with an increase in hemoglobin concentration among individuals with normal or mildly decreased renal function [31]. In the present study, the change in hemoglobin concentration was inversely correlated with the eGFR after 1 year among members of the general population with preserved renal function, consistent with the previous findings [31]. Further studies are needed to elucidate the pathogenesis involved in the inverse relationship between the change in renal function and the change in hemoglobin concentration among members of the general population.

In living kidney donors, the GFR declines with increasing age [32]. An observational study involving the general population indicated that eGFR decreased as age increased [22]. In the present study, age was negatively associated with the eGFR after 1 year among members of the general population. These findings suggest that age is negatively associated with changes in renal function in the population both with and without CKD.

In the general population, there is evidence that eGFR decline accelerates as eGFR decreases [33]. In the present study, eGFR was positively associated with the eGFR after 1 year among members of the general population. These results suggest that current renal function data are essential for predicting subsequent changes in renal function among members of the general population.

A large population-based cohort study involving 120,727 individuals in Japan reported that the prevalences of CKD G3 and G4-5 were 21.35% and 0.01%, respectively [33]. In the present study, the prevalences of CKD G3, G4, and G5 were 14.6%, 0.1%, and 0.0%, respectively, which was similar to the result of previous study [33].

This study had two main advantages. First, it was a large-scale cohort study involving approximately 40,000 individuals in the general population, and it analyzed clinical parameters associated with the eGFR after 1 year. We confirmed that age, hemoglobin, uric acid, and eGFR are associated with a future change in eGFR, as reported in a population with CKD. The results of our study can facilitate further research to identify clinical factors associated with future change in eGFR among members of the general population. Second, to our knowledge, this is the first study to develop an approximation formula for predicting GFR after 1 year using clinical parameters associated with the eGFR after 1 year. The approximation formula derived from age, hemoglobin, uric acid, and eGFR may be useful in the prediction of GFR after 1 year among members of the general population.

This study also had some limitations. First, its retrospective design might have led to some reporting and selection biases. Second, all participants were residents of Japan, which might reduce the generalizability of the findings. Additional prospective studies involving multiethnic populations are needed to confirm our findings. Third, we used a two-point method to calculate the annual changes in clinical parameters including hemoglobin and uric acid, which may have introduced larger variability because it does not contain any information between the two points [34]. Fourth, we used the random forest method as a machine learning algorithm because it is superior with respect to high accuracy, good performance with many variables, and resistance to a large amount of learning data [35]. However, the conclusion can hardly be stronger than the stringency of the data which entered the machine learning algorithm. Fifth, in the present study, we developed an approximation formula for predicting GFR after 1 year using eGFR after 1 year calculated by the Modification of Diet in Renal Disease formula as a reference standard. However, it has been shown that variation between GFR measured by inulin clearance and eGFR calculated by the Modification of Diet in Renal Disease formula was greater in the population without CKD [23]. The possibility remains that our formula may not accurately approximate GFR after 1 year among members of the general population. Sixth, in the present study, we did not analyze the clinical parameters associated with the annual change in eGFR. Therefore, further studies are required to assess the accuracy of the approximation formula derived from clinical parameters associated with the annual change in eGFR, in comparison with that of the approximation formula derived from clinical parameters associated with the eGFR after 1 year.

In conclusion, age, hemoglobin, uric acid, and eGFR were associated with the eGFR after 1 year among members of the general population in Japan. An approximation formula including age, hemoglobin, uric acid, and eGFR may be useful for predicting GFR after 1 year in the general population. However, there was a non-negligible discrepancy between the approximate GFR after 1 year and the eGFR after 1 year. This study provides the foundation for additional research to develop an approximation formula for predicting GFR in the general population.

## Figures and Tables

**Figure 1 jcm-13-04207-f001:**
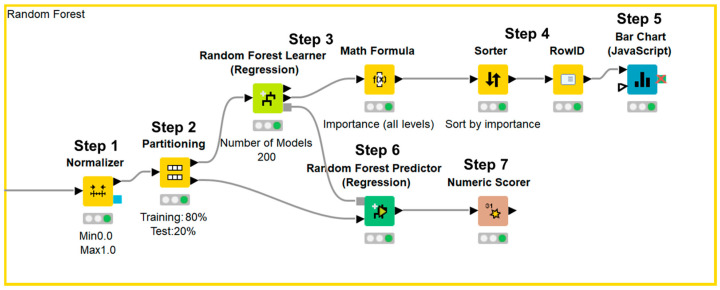
KNIME workflow for random forest method.

**Figure 2 jcm-13-04207-f002:**
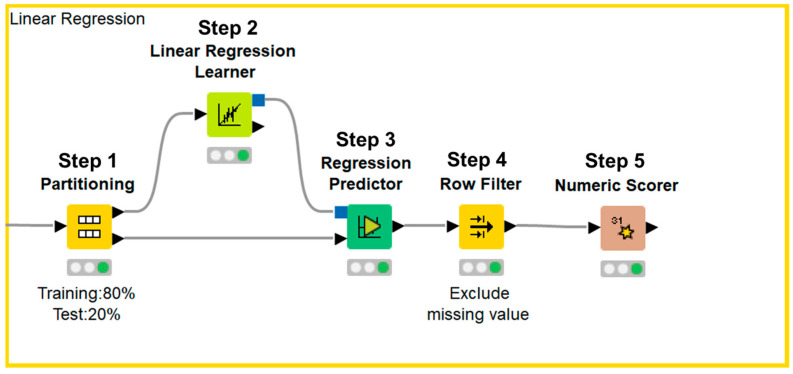
KNIME workflow for multiple linear regression analysis.

**Figure 3 jcm-13-04207-f003:**
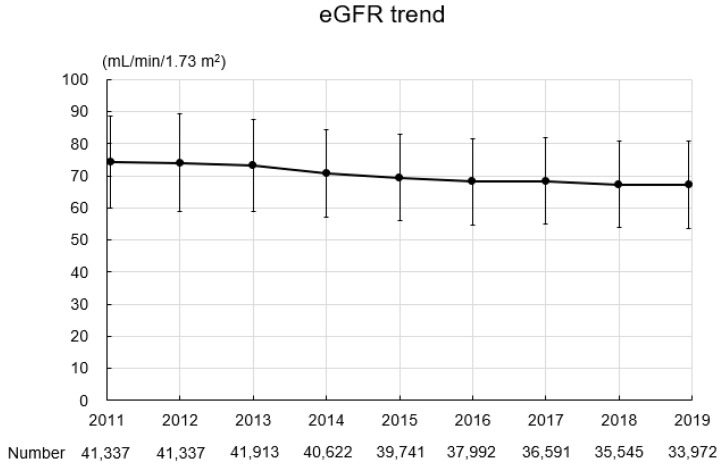
eGFR trend during the study period. Abbreviation: eGFR, estimated glomerular filtration rate.

**Figure 4 jcm-13-04207-f004:**
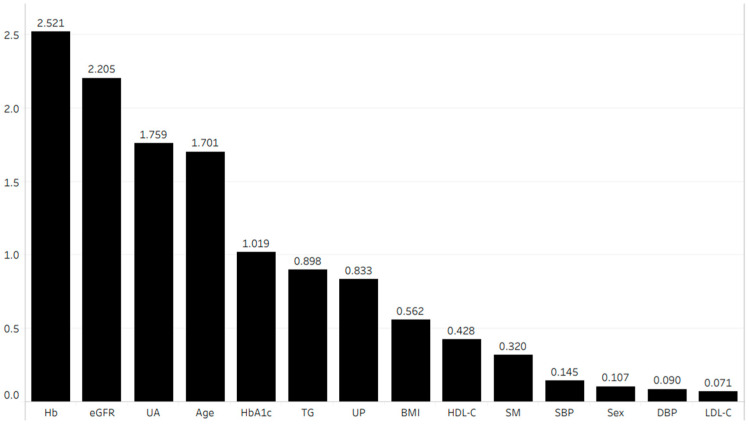
Importance of each parameter in terms of its association with eGFR after 1 year. Abbreviations: BMI, body mass index; DBP, diastolic blood pressure; eGFR, estimated glomerular filtration rate; Hb, hemoglobin; HbA1c, hemoglobin A1c; HDL-C, high-density lipoprotein cholesterol; LDL-C, low-density lipoprotein cholesterol; SBP, systolic blood pressure; SM, smoking history; TG, triglyceride; UA, uric acid; UP, urinary protein.

**Figure 5 jcm-13-04207-f005:**
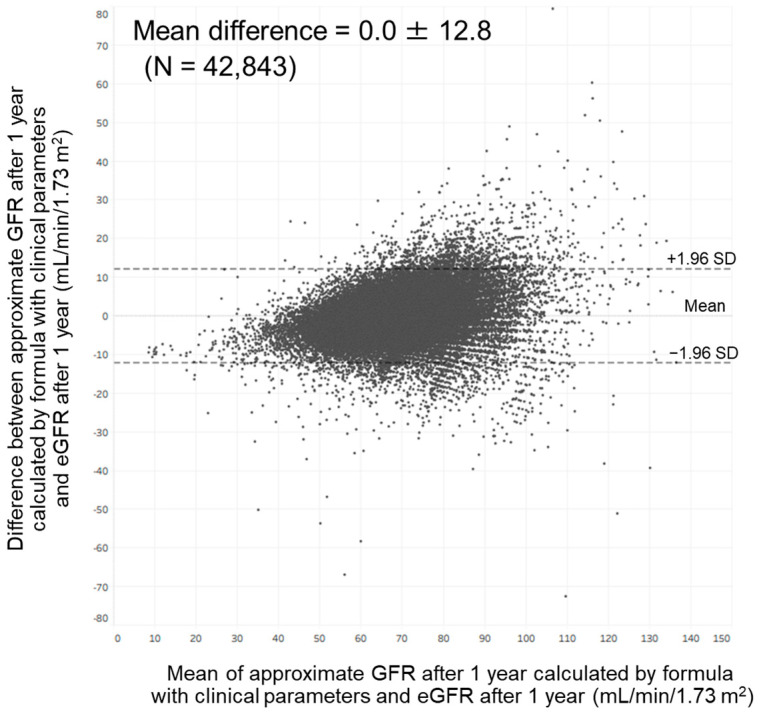
Bland–Altman plot comparing approximate GFR after 1 year calculated by our formula with clinical parameters and estimated GFR after 1 year. Abbreviations: eGFR, estimated glomerular filtration rate; GFR, glomerular filtration rate; SD: standard deviation.

**Table 1 jcm-13-04207-t001:** Studies that investigated the factors associated with renal function decline.

Risk Factor	Author (Reference)	Year	Risk Factor	Author (Reference)	Year
Male sex	Swartling et al. [4]	2021	Hemoglobin	Yang et al. [12]	2020
	Chesnaye et al. [5]	2021		Pan et al. [13]	2022
Age	Rule et al. [6]	2010	Uric acid	Tseng et al. [14]	2019
Smoking	Choi et al. [7]	2019		Xiong et al. [15]	2022
	Ito et al. [8]	2024	Triglyceride	Zhang et al. [16]	2019
BMI	Hung et al. [9]	2022	HDL-C	Rahman et al. [17]	2014
SBP	Lee et al. [10]	2020	LDL-C	Goeij et al. [18]	2015
	Lee et al. [11]	2021	HbA1c	Jiang et al. [19]	2019
DBP	Lee et al. [10]	2020		Moriya et al. [20]	2022
	Lee et al. [11]	2021	Urinary protein	Su et al. [21]	2020
			eGFR	Baba et al. [22]	2015

Abbreviations; BMI, body mass index; DBP, diastolic blood pressure; eGFR, estimated glomerular filtration rate; HbA1c, hemoglobin A1c; HDL-C, high-density lipoprotein cholesterol; LDL-C, low-density lipoprotein cholesterol; SBP, systolic blood pressure.

**Table 2 jcm-13-04207-t002:** Baseline participant characteristics.

Characteristic		All participants (*n* = 41,337)
Men (number, %)		16,918 (40.9)
Age (years)		64.0 ± 6.9
Smoking (number, %)		4561 (11.0)
Body mass index (kg/m^2^)		22.8 ± 3.2
Systolic blood pressure (mmHg)		128.2 ± 16.2
Diastolic blood pressure (mmHg)		76.3 ± 10.6
Hemoglobin (g/dL)		13.8 ± 1.3
Uric acid (mg/dL)		5.2 ± 1.3
Triglyceride (mg/dL)		108.9 ± 68.0
High-density lipoprotein cholesterol (mg/dL)		63.8 ± 15.8
Low-density lipoprotein cholesterol (mg/dL)		126.2 ± 30.0
Hemoglobin A1c (%)		5.9 ± 0.6
Urinary protein grade (number, %)	1: −	35,542 (86.0)
	2: ±	3648 (8.8)
	3: 1+	1651 (4.0)
	4: 2+	397 (1.0)
	5: ≥3+	96 (0.2)
Estimated glomerular filtration rate (mL/min/1.73 m^2^)		74.3 ± 14.2
Estimated glomerular filtration rate category (number, %)	G1: ≥90 mL/min/1.73 m^2^	28,198 (68.2)
	G2: 60–89 mL/min/1.73 m^2^	7037 (17.0)
	G3a: 45–59 mL/min/1.73 m^2^	5506 (13.3)
	G3b: 30–44 mL/min/1.73 m^2^	525 (1.3)
	G4: 15–29 mL/min/1.73 m^2^	54 (0.1)
	G5: <15 mL/min/1.73 m^2^	17 (0.0)

Data are presented as mean ± standard deviation or number (%).

**Table 3 jcm-13-04207-t003:** Multiple linear regression analysis of variables correlated with the eGFR after 1 year.

**(A) Using clinical parameters as independent variables**		
Variable	Coefficient	*p*-value
Age (years)	−0.054	<0.001
Hemoglobin (g/dL)	0.162	<0.001
Uric acid (mg/dL)	−0.085	<0.001
Estimated glomerular filtration rate (mL/min/1.73 m^2^)	0.849	<0.001
Intercept	11.5	<0.001
**(B) Using annual changes in clinical parameters as independent variables**		
Variable	Coefficient	*p*-value
Age (years)	−0.050	<0.001
Annual change in hemoglobin (g/dL)	−0.398	<0.001
Annual change in uric acid (mg/dL)	−3.205	<0.001
Estimated glomerular filtration rate (mL/min/1.73 m^2^)	0.864	<0.001
Intercept	11.9	<0.001

Coefficients represent the changes in the dependent variable per unit changes in independent variables.

## Data Availability

Datasets used in the current study are available from the corresponding author on reasonable request.

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
