# Peer review of "Approximation of Glomerular Filtration Rate after 1 Year Using Annual Medical Examination Data"

_jcm, 2024, doi:10.3390/jcm13144207_

Round 1
Reviewer 1 Report (Previous Reviewer 3)
Comments and Suggestions for Authors
This report is now much better than the previous report. I believe that it can be condensed in some areas and there are a few significant issues that should be addressed.
Significant comments;
1. Table 1 has to be much shorter, and I suggest considering leaving out several references that are redundant. Table 1 could include only the Risk Factor and the references related to it. For example, the 1st row would list "Male sex" in column 1 and references 4 - 8 in a 2nd column. Then "Age" and references 9 - 10. Etc.
The number of references for each Risk Factor seem excessive and could be shortened to still make the case. For example, Uric Acid could list only 2 or 3 references instead of 6.
2. I do not believe Figure 5 is needed because Figure 6 gives the information more clearly.
3. Given that Figure 3 shows an average yearly decline in eGFR of approximately 0.88 mL/min with age (as expected), I would like to see how close the eGFR calculated using that -0.88 mL/min factor approximates the measured eGFR after 1 year, in comparison to how close the eGFR by the approximation equation agrees with the measured eGFR. For example, if a patient's measured eGFR declines by 2 mL/min in a year and the equation predicts a decline of 4 mL/min, the yearly factor would be off by 1.12, while the equation would be off by 2 mL/min.
Minor comments:
Abstract, Results and elsewhere: Please do not use excessive significant figures in the numbers. For example, the approximation equation used the factor 11.476. Given that this is an eGFR, I suggest this should be rounded off to 11.5.
page 6, 1st paragraph: Given that blood pressure is a minor contributor (as shown in Fig 4) I suggest giving less detail on how blood pressure was measured.
Page 11, next to last paragraph, 1st sentence: I suggest deleting ...""prevalence of CKD G 3-5 was 21.36% and the"... . The last part of that sentence is adequate.
Author Response
Responses to Reviewer #1
Comment 1: Table 1 has to be much shorter, and I suggest considering leaving out several references that are redundant. Table 1 could include only the Risk Factor and the references related to it. For example, the 1st row would list "Male sex" in column 1 and references 4 - 8 in a 2nd column. Then "Age" and references 9 - 10. Etc. The number of references for each Risk Factor seem excessive and could be shortened to still make the case. For example, Uric Acid could list only 2 or 3 references instead of 6.
In accordance with the reviewer’s comment, we have modified Table 1 by reducing rows and columns.
Comment 2: I do not believe Figure 5 is needed because Figure 6 gives the information more clearly.
In accordance with the reviewer’s comment, we have removed Figure 5 from the revised manuscript.
Comment 3: Given that Figure 3 shows an average yearly decline in eGFR of approximately 0.88 mL/min with age (as expected), I would like to see how close the eGFR calculated using that -0.88 mL/min factor approximates the measured eGFR after 1 year, in comparison to how close the eGFR by the approximation equation agrees with the measured eGFR. For example, if a patient's measured eGFR declines by 2 mL/min in a year and the equation predicts a decline of 4 mL/min, the yearly factor would be off by 1.12, while the equation would be off by 2 mL/min.
Thank you for your comment. As you noted, in the present study, we did not analyze the clinical parameters associated with the annual change in eGFR. Therefore, based on the results of the present study, we expect to perform further studies to assess the accuracy of the approximation formula derived from clinical parameters associated with the annual change in eGFR, in comparison with that of the approximation formula derived from clinical parameters associated with the eGFR after 1 year. We have added the following description to the Discussion section.
“Sixth, in the present study, we did not analyze the clinical parameters associated with the annual change in eGFR. Therefore, further studies are required to assess the accuracy of the approximation formula derived from clinical parameters associated with the annual change in eGFR, in comparison with that of the approximation formula derived from clinical parameters associated with the eGFR after 1 year.”
Comment 4: Abstract, Results and elsewhere: Please do not use excessive significant figures in the numbers. For example, the approximation equation used the factor 11.476. Given that this is an eGFR, I suggest this should be rounded off to 11.5.
Thank you for your comment. We have rounded off 11.476 to 11.5 throughout the revised manuscript. We have also rounded off 11.931 to 11.9 throughout the revised manuscript.
Comment 5: page 6, 1st paragraph: Given that blood pressure is a minor contributor (as shown in Fig 4) I suggest giving less detail on how blood pressure was measured.
In accordance with the reviewer’s comment, we have revised the descriptions regarding how blood pressure was measured, as follows.
“Blood pressure was measured with the participant in a sitting position at rest, using an automated upper arm cuff.”
Comment 6: Page 11, next to last paragraph, 1st sentence: I suggest deleting ...""prevalence of CKD G 3-5 was 21.36% and the"... . The last part of that sentence is adequate.
In accordance with the reviewer’s comment, we have deleted the phrase "prevalence of CKD G3-5 was 21.36% and the" from the sentence.
Reviewer 2 Report (Previous Reviewer 1)
Comments and Suggestions for Authors
We are dealing with the resubmitted version of a manuscript which I reviewed in the past. The reviewer feels the authors have made efforts to considerably improve the quality of the resubmitted manuscript. Most of the comments/questions have been tackled in the resubmitted version. Although the quality has improved it will remain true that the approximations/algorithms generated/suggested by machine learning using the random forest method will critically depend on the input of mainly participants from the general populations for which eGFR algorithms are poor measures versus mGFR if the participants do not have CKD. Nonetheless the reviewer appreciates the authors' exercise trying to "predict" GFR after 1 year using "elementary" clinical data from yearly annual examinations. The reviewer does not have further questions/comments.
Author Response
Responses to Reviewer#2
Comment 1: We are dealing with the resubmitted version of a manuscript which I reviewed in the past. The reviewer feels the authors have made efforts to considerably improve the quality of the resubmitted manuscript. Most of the comments/questions have been tackled in the resubmitted version. Although the quality has improved it will remain true that the approximations/algorithms generated/suggested by machine learning using the random forest method will critically depend on the input of mainly participants from the general populations for which eGFR algorithms are poor measures versus mGFR if the participants do not have CKD. Nonetheless the reviewer appreciates the authors' exercise trying to "predict" GFR after 1 year using "elementary" clinical data from yearly annual examinations. The reviewer does not have further questions/comments.
We thank the reviewer for the positive comments on our manuscript.
This manuscript is a resubmission of an earlier submission. The following is a list of the peer review reports and author responses from that submission.
Round 1
Reviewer 1 Report
Comments and Suggestions for Authors
JCM-2964676 Hirai et al
Approximation of estimated glomerular filtration rate after 1 year using annual medical examination data
Summary
The authors created two approximation formulas for GFR after 1 year by a machine learning –based random forest method (KNIME Analytics Platform) which was fed by consecutive annual medical examination data (period 2011-2019) from >40, 000 individuals from Saitama City (primary care and community hospitals). The final dataset include 14 variables: sex, age, smoking, BMI, SBP, DBP, Hb, UA, TG, HDL-C, LDL-C, HbA1c, UP (grades) and eGFR. Machine learning identified 4 major explaining baseline parameters (eq. 1: age, Hb, UA, eGFR) for a first approximation formula of GFR after one year and 4 parameters (eq. 2: age, eGFR and annual changes of Hb and UA) for the second approximation of GFR after one year. Beta coefficients were obtained from multiple linear regression analyses. eGFR was calculated by the 2009 MDRD formula for Japan. The correlation between approximate eGFR after 1 year and eGFR after 1 year was strong for both equations (r > 0.85). In Bland-Altman plots the mean differences between “approximate GFR after 1 year” and “eGFR after 1 year” were extremely small (0.0) but 1.96SD limits were not optimal. The authors concluded that the approximation formula with age, Hb, UA and eGFR might be useful for prediction of eGFR after 1 year in the general population.
Comments
1. The authors claim novelty for both their approximation formulas for GFR after one year. Indeed those data are novel information. It is the first time that approximation formulas for GFR (based on age, current eGFR, UA and Hb) after one year have been launched. However the reviewer is far from optimistic when reading their data. See comments below.
2. Indeed machine learning by the random forest method fed by a large amount (> 40 000 individuals) of data from a population-based cohort theoretically should allow for strong prediction data. However the weakness of machine learning is that the conclusion can hardly be stronger than the stringency of the data which entered the machine learning algorithms. And that is the major problem with the analyses. The reviewer is not at all surprised by the excellent correlations with eGFR (r > 0.85) because the machine learning of course looks for best approximations with eGFR. The basic problem is that eGFR in the general population is a poor measure of glomerular filtration rate and both approximation formulas are best approximations for a poor measure of glomerular filtration rate in the general population. See comment below.
3. The authors used (Ref 10) the old formula for Japanese individuals (AJKD 2009) to estimate GFR. It is a modification of the MDRD formula to estimate GFR. In the mean time dozens of papers have been published that MDRD-based algorithms perform very well for CKD patients (eGFR <60 ml/min/1.73 m2) but very poorly in the general population without CKD (> 60 ml/min/ 1.73m2). Errors are considerable (Sometimes > 10%). Thus the validity of the approximation formulas for the GENERAL population despite large numbers and machine learning techniques remains questionable. Unfortunately the authors did not even discuss that major problem.
4. The authors should refrain from using the wording “measured GFR” as it is misleading. GFR has not been measured (mGFR is a specific methodology e.g. iohexol clearance and so on). They compared eGFR after one year (Japanese MDRD) with approximate GFR after one year (machine learning and regression models).
5. The authors should better specify the a priori selection of the 14 variables. Was it based on the literature, based on the available parameters in Saitama City or based on best educated guesses? It is striking that there is only one urinary parameter: UP (based on grades). Was there a kind of quality control on those parameters before entering the machine learning algorithms? Are not the authors afraid of large errors for annual changes in uric acid and Hb when applying it to the general population?
6. Statistics in Tables 2 and 3. Are the beta-coefficients standardized beta-coefficients or expressed per unit change?
7. Fig 8. The reviewer might be wrong but at first glance the B-A plots look trended. Is it right?
8. Discussion. As mentioned above the authors do not discuss the major limitations of their study but only some minor points.
Comments on the Quality of English LanguageOnly minor editing required.
Reviewer 2 Report
Comments and Suggestions for Authors
Dear authors,
I want to congratulate the authors for the quality of the paper they provided.
The objectives and methodology are clearly defined. The results that have been presented are relevant. The bibliography is up to date.
To enhance the manuscript, some minor questions need to be addressed: the
introduction requires more details.
Comment 1. Discuss the prevalence of chronic kidney disease in Japan.
Comment 2. It is necessary to thoroughly review existing literature and studies that explore these correlations in the glomerular filtration rate. The data from these studies is valuable in understanding the dynamics of kidney function in relation to these variables in the Japanese population. The authors must cite studies and present the topic in question.Consequently, the justification for the study will improve.
Kind regards.
Reviewer 3 Report
Comments and Suggestions for Authors
This is an interesting approach to predicting future eGFRs. They measured many parameters and concluded that the eGFR, Hb, Uric Acid, and age could be entered into an equation to predict the 1 year follow up of eGFR.
My concern is with the practical clinical value of using these markers to predict the likelihood of the eGFR in 1 year based on population data. Figure 8 shows there is a lot of scatter. However, this report could be much more useful by developing an equation that will predict the likelihood of an individual person having a 1-year decrease in eGFR of <5, 10-15, 15-20, or 20+ (as examples). For example, the equation could tell a person they have a >50% likelihood their eGFR will decrease by more than 10 mL/min in 1 year. Thus, this could predict if a specific patient is likely to have a significant deterioration of kidney function in the next year.
Other comments:
This report is of some interest in confirming earlier studies that age, Hb, US, and eGFR are associated with the future change of eGFR.
Discussion: It is not too surprising that these parameters have similar predictors in both non-CKD persons and persons with CKD.
Tables 2 and 3 could be eliminated by listing Coefficients and P-values in the text. At least combine them into 1 table.
I do not see that Figure 1 adds anything to the report.
I cannot understand Figures 3 or 4. I suggest a statistician review for clarity and relevance.
Figures 4 and 5 are good. Not sure that Figure 6 adds much to the report.
For Figures 7 and 8, I do not see that showing how the annual change predicts the annual change is useful. Maybe consider using just Figure 7a and 8a.